Five decades of change in somatic growth of Pacific hake from Puget Sound and Strait of Georgia

Chittaro Paul paul.chittaro@noaa.gov 1
Grandin Chris 2
Pacunski Robert 3
Zabel Rich 4
1 Environmental and Fisheries Sciences Division, Northwest Fisheries Science Center, National Oceanic and Atmospheric Administration , Seattle , WA , United States of America
2 Marine Ecosystem and Aquaculture Division, Pacific Biological Station, Fisheries and Oceans Canada , Nanaimo , British Columbia , Canada
3 Washington Department of Fish and Wildlife , Olympia , WA , United States of America
4 Fish Ecology Division, Northwest Fisheries Science Center, National Oceanic and Atmospheric Administration , Seattle , WA , United States of America
Zucchetta Matteo
Electronic publication date: 2022 Jul 13
Publication date: 2022
Volume: 10
Electronic Location ID: e13577
Received 2021 Dec 22; Accepted 2022 May 22
Copyright: ©2022 Chittaro et al.
Copyright year: 2022
Copyright holder: Chittaro et al.
License: This is an open access article distributed under the terms of the Creative Commons Attribution License, which permits unrestricted use, distribution, reproduction and adaptation in any medium and for any purpose provided that it is properly attributed. For attribution, the original author(s), title, publication source (PeerJ) and either DOI or URL of the article must be cited.
License URL: https://creativecommons.org/licenses/by/4.0/

Keywords: Species of concern, Trace elements, Fish body size, Pacific whiting, Population recovery

Funding: The authors received no funding for this work.

==============================
Declines in fish body size have been reported in many populations and these changes likely have important ramifications for the sustainability of harvested species and ecosystem function. Pacific hake, Merluccius productus, have shown declines in size over the last several decades for populations located in Puget Sound (PS), Washington, USA, and Strait of Georgia (SoG), British Columbia, Canada. To examine this decrease in size, we used archived otoliths from both populations to assess when the decrease in somatic growth occurred and explored what factors and processes might explain the decline, including otolith microchemistry to infer the environment experienced by fish at different ages. Results indicated that substantial changes in juvenile somatic growth have occurred across decades. The divergence in body size occurred in the second summer, whereby SoG fish grew, on average, 18% more than PS fish. Within the PS population, somatic growth differed significantly among fish that hatched in the 1980s, 1990s, and 2010s, such that the more recently hatched fish grew 26% more in their first summer and 71% less in their second summer relative to those that hatched in the 1980s. In comparison, growth of SoG fish did not differ between those that hatched in 1970s and 1990s. For both populations growth in the first and third summer was positively and negatively related, respectively, to the abundance of harbor seals, while growth in the first and second summer was negatively related to salinity. Overall, this study highlights the complicated nature of Pacific hake population recovery under dynamic, and typically uncontrollable, variation in biotic and abiotic conditions.

Introduction

Some populations respond to mitigation actions, such as harvest regulation, while other populations do not. Atlantic cod (Gadus morhua), for example, has not recovered from decades of overfishing despite harvest being largely curtailed (Olsen et al., 2004). This is in contrast to species recovering and supporting sustainable fisheries around the world (Hilborn et al., 2020) some of which are in U.S. waters (NOAA Fisheries, 2020). For instance, on the U.S. west coast the population of Pacific Hake (Merluccius productus), which ranges from southern California to northern British Columbia, Canada, has responded quite favorably to harvest regulation and is the largest fishery (by metric tons landed) on the U.S. West Coast (Grandin et al., 2020). Nonetheless, a closely related population, found in Puget Sound, Washington, USA has been subjected to fishing regulations (i.e., fishery closure) since 1991, but still has not recovered (Gustafson et al., 2000).

The Puget Sound (PS) population along with the Strait of Georgia (SoG) population, in British Columbia, Canada, are considered a distinct population segment (DPS)1 under the U.S. Endangered Species Act (ESA). This DPS, referred to as the Georgia Basin DPS (Fig. 1), was listed as a Species of Concern under the ESA by both U.S. federal and state agencies, in part due to the decades-long declining status of the PS population (Gustafson et al., 2000; WDFW, 2008).

Figure 1 Map of Strait of Georgia (Canada) and Puget Sound (USA) where Pacific hake (Merluccius productus) were collected.

White ovals indicate location of known and historical spawning ground. Black circles indicate collection sites at (A) Lasqueti Island, (B) Gabriola Island, (C) Burrard Inlet, and (D) Port Susan.

In addition to a decrease in biomass of approximately 85% since at least the 1980s, the PS population has also experienced substantial declines in body size (26% for 5- to 7-year-olds) during this same time period (Gustafson et al., 2000), some of which is attributable to high harvest rates in the early and mid-1980s (Goñi, 1988). In contrast, the SoG population has experienced moderate reductions in size (approximately 12% reduction for 4 and 5 year olds) and biomass (35%) from 1981 to 1997 (King & McFarlane, 2006). Taken together, these declines in the Georgia Basin DPS biomass and body size highlight that changes to the prey and/or habitat resources or processes (e.g., predation) critical to a specific life stage (e.g., juveniles) have occurred and may have impacted the sustainability of the DPS.

In this study we evaluate Pacific hake resource quality and availability in both the PS and SoG populations by quantifying somatic growth in individuals that hatched over a 41-year period (1974–2015). We used somatic growth as an indicator of individual performance and habitat quality (Le Pape et al., 2003; Meng et al., 2000; Necaise, Ross & Miller, 2005) given the known positive relationships between body size and fitness (Kingsolver & Huey, 2008), and survival (Juan-Jordá et al., 2015; Sogard, 1997; Zabel & Achord, 2004). Measuring somatic growth offers a direct method of evaluating the physiological status (i.e., performance) of an individual because somatic growth is related to factors such as food quality (Steves & Cowen, 2000) and availability (Graeb et al., 2004; Ligas et al., 2015), temperature (Baumann et al., 2006; Ligas et al., 2015), and processes like competition (Helser & Almeida, 1997) and predation (Baumann et al., 2003). To reconstruct somatic growth of an individual we counted visible increments in their otoliths to estimate age (Jones, 1992; Stevenson & Campana, 1992) and used body size-age relationships to estimate individual somatic growth during each summer and winter (Casselman, 1987). Using this measure of performance, we evaluated the extent to which size-at-age varied within and between populations, how size-at-age varied across years, and what factors and/or processes putatively explain variability in somatic growth.

Materials and methods

Study species

Pacific hake (Merluccius productus) is a pelagic species distributed from Baja California, USA to northern B.C., Canada (Berger et al., 2019). We studied size-at-age patterns of Pacific hake within the Georgia Basin DPS, which includes individuals from populations in greater Puget Sound (PS) and the Strait of Georgia (SoG) (Fig. 1) for which genetic differences have been reported (Iwamoto, Ford & Gustafson, 2004; Iwamoto et al., 2015). In PS, spawning aggregations occur in Port Susan (Pedersen, 1985) and Dabob Bay (Utter, Stormont & Hodgins, 1970). In SoG, spawning aggregations occur in Stuart Channel (McFarlane & Beamish, 1985) and Saanich Inlet (Beamish, Smith & Scarsbrook, 1978). Individuals mature by age 3–4 (McFarlane & Beamish, 1985) and spawning occurs primarily from February through April, peaking in March (Goñi, 1988). Hatching takes place 4–6 days after fertilization, at which point larvae are 2–3 mm total length (Bailey, 1982b; Hollowed, 1992).

Fish collections and otolith preparation

Using pelagic trawls, juvenile and adult Pacific hake were collected from PS and SoG spawning grounds by the Washington Department of Fish and Wildlife (WDFW) (Burger et al., 2019) and Fisheries and Oceans, Canada (DFO), respectively (Fig. 1) across several years (Table 1). Fish total length (TL) was measured, and otoliths were removed and stored dry. Left sagittal otoliths were mounted on glass microscope slides using thermoplastic cement; otoliths were then polished in a sagittal plane using a grinding wheel with slurries of 600-grit silicon carbide, 5.0 alumina oxide, and 1.0 micropolish (Buehler2) until the core was visible. Images of each otolith were taken with transmitted light using a digital camera (Mediacybernetics, EvolutionMP) at 4 × magnification and analyzed using image processing software (Image Pro Plus, version 7.0).

Table 1 Number of sampled Pacific hake per population, year, and age.

	Strait of Georgia		Puget Sound			
Year\age	2	3	4	5	6		0	1	2	3	4	5	6		Total	
1979		7	16	6											29	
1982		8	19	4											31	
1987									5	18	2				25	
1996	12								27						39	
1997	13								35	1					49	
1999	23	7													30	
2000									4	72	7	1			84	
2001									27	8					35	
2002			2	12	4		19	8	2	8	8	10	1		74	
2005							3								3	
2008							2								2	
2016								3	22	10	3				38	
Total	48	22	37	22	4	133	24	11	122	117	20	11	1	306	439	

Figure 2 Pacific hake otolith microstructure and microchemistry.

(A) Image of a Pacific hake otolith (individual WDFW-2000-194 under transmitted light) showing the opaque and translucent zones that correspond to summer and winter otolith growth, respectively for each of three years. (B) Calcium intensity (counts per second, cps; yr1S and yr1W corresponds to year one summer and year one winter, respectively) and ratios of magnesium, manganese, strontium, barium, and lead (Mg, Mn, Sr, Ba, Pb, respectively) to calcium (Ca) (mm mol−1) with respect to distance from the otolith core (microns) as measured by laser ablation inductively coupled plasma mass spectrometry.

Age, length, and growth

To estimate age and change in size over time, we examined the annual pattern of otolith growth, which consists of an opaque and translucent zone (Chilton & Beamish, 1982). Under transmitted light, the opaque zone corresponds to a summertime period when fish do most of their growing, while the translucent zone is a period of reduced winter growth (Beamish, 1979; Chilton & Beamish, 1982) (Fig. 2A). For each fish, we estimated age by counting the number of otolith translucent zones (Bailey, 1982b; Beamish, 1979) along a transect perpendicular to its longest axis on the dorsal side. Age was estimated for PS and SoG fish by WDFW in Olympia, Washington, and by DFO at the Pacific Biological Station in Nanaimo, BC, respectively. A second age was estimated by the U.S. National Marine Fisheries Service, Northwest Fisheries Science Center in Seattle, Washington. If the two ages differed for an individual, then age was estimated a third and final time by the Northwest Fisheries Science Center. Fish were assigned an age and the otolith included for statistical analysis only if two age estimates were the same.

Because somatic and otolith growth are positively correlated and consistent between both populations (R2 = 0.83, n = 532, Fig. S1) we used the otolith radius at opaque and translucent zones to estimate fish length at the end of each summer and winter, respectively. Specifically, to back-calculate summer and winter body sizes we measured the otolith radius to each opaque (Os) and translucent (Ow) zone, as well as the distance from the otolith core to the edge (i.e., otolith radius at time of capture, Oc) (Beamish, 1979) (Fig. 2A). For each individual, fish length after each summer (Ls) and winter (Lw) was estimated using the Biological Intercept equation (Campana, 2011; Campana & Jones, 1992): Ls=Lc+Os−OcxLc−LiOc−Oi

where Li and Oi were the biological intercepts and were defined as 3.9 mm and 14.1 µm, respectively and Lc corresponds to the length at capture. The biological intercepts corresponded to the size at first feeding and were based on work by Bailey (1982a) and Butler & Nishimoto (1997). In addition, we calculated the estimated amount an individual grew (mm) each summer and winter as the difference between its length at any summer or winter from its length estimated for the previous winter or summer, respectively.

Our statistical analyses of length and growth data had two objectives. Our first objective was to identify when size differences arose between PS and SoG fish. We used a repeated measures ANOVA to compare length and growth between fish collected from PS and SoG, regardless of hatch year. Repeated measures ANOVA was appropriate because our otolith-derived estimates of length and growth represent repeated measures that were calculated for each age-season of a fish’s life (i.e., first summer, first winter, second summer, etc.) (Table 2).

Table 2 Number of otolith-derived length estimates per age (1–5), season (summer or winter), population (Strait of Georgia, SoG, or Puget Sound, PS), and hatch year.

Most fish were greater than 1 year old and so otoliths were often used to estimate length at several years and seasons.

Age	1	1	2	2	3	3	4	4	5	5	
Season	Summer	Winter	Summer	Winter	Summer	Winter	Summer	Winter	Summer	Winter	
Hatch year	SoG	PS	SoG	PS	SoG	PS	SoG	PS	SoG	PS	SoG	PS	SoG	PS	SoG	PS	SoG	PS	SoG	PS	
1974	6		6		6		6		6		6		6		6		6		6		
1975	16		16		16		16		16		16		16		16						
1976	7		7		7		7		7		7										
1977	4		4		4		4		4		4		4		4		4		4		
1978	19		19		19		19		19		19		19		18						
1979	8		8		8		8		8		8										
1983		2		2		2		2		2		2		2		2					
1984		21		18		18		18		18		18									
1985		10		5		5		5													
1994	12	30	12	28	12	28	12	28		1		1									
1995	13	36	13	36	13	36	13	36		1		1		1		1		1		1	
1996	11	8	11	8	11	8	11	8	11	8	11	8	4	7	4	7	4	1	4	1	
1997	35	83	35	82	35	82	35	82	12	82	12	82	12	1	12	1	12	1	12	1	
1998	2	21	2	20	2	20	2	20	2	16	2	16	2	8	2	8					
1999		36		35		35		35		8		8									
2000		3		2		2		2													
2001		8		8																	
2002		25																			
2005		3																			
2008		2																			
2012		3		3		3		3		3		3		3		3					
2013		10		10		10		10		10		10									
2014		22		22		22		22													
2015		3		3																	
Total	133	306	135	284	134	272	134	272	85	149	85	149	63	31	62	31	26	12	26	12	

The second objective was to determine the extent to which length and growth have changed across hatch years, grouped by decade, within a population. We used a repeated measures ANOVA to investigate if length and growth differed among each year/season and among hatch years. We grouped fish according to the decade in which they hatched because low sample sizes for certain hatch years precluded year-specific analysis. If significant differences were detected we performed a Tukey’s post-hoc test to determine which decades were significantly different in terms of length and growth. We used a mixed effects model to describe the repeated measures analysis using the lme function in the nlme package of RStudio (version 1.1.463) (R Core Team, 2021). ‘Individual’ was treated as a random variable in the model and we included an autocorrelation structure for lags in the time variable (i.e., year/season) via the ACF function in the nlme package.

Otolith chemistry

We chemically analyzed the same otoliths from which we estimated age and seasonal growth since ratios of certain isotopes are known to reflect the environment in which the fish resides (Elsdon et al., 2008). Prior to their chemical analysis, otoliths mounted to microscope slides were cleaned by rinsing with 95% ethanol. We used laser ablation inductively coupled plasma mass spectrometry (LA-ICP-MS) at the GeoAnalytical Lab, Washington State University. Each laser ablation of an otolith obtained trace elemental concentrations from the otolith edge, which corresponds in space and time to their collection from a spawning ground, to the otolith core, which corresponds to hatching.

Data were collected as previously described in Chittaro et al. (2013). Specifically, isotope concentrations were determined using high resolution single collector inductively coupled plasma mass spectrometry (Finnigan Element2, with helium as the carrier gas) that was coupled with a laser ablation system (New Wave UP-213, frequency of 20-Hz, 30-µm spot size). Using an automated microscope stage, the laser beam was focused on the otolith, and a transect from edge to its core was ablated at a speed of 15 µm s−1. Each transect was placed along the same axis used to estimate age and size, and formed an edge-to-core scan line corresponding to the entire life of the fish. Data acquisition of the LA-ICP-MS lasted 240 s, 20 s of which were designated for instrument calibration and gas background counts prior to the start of each ablation. To correct for instrument drift, we obtained a glass standard doped with trace elements from the National Institute of Standards and Technology (NIST 610). We analyzed this standard both at the beginning and end of each sample set (i.e., 16–20 otoliths). Calcium was used as an internal standard to compensate for signal variation caused by differences in the mass of ablated material. We calculated detection limits for each isotope as the average background plus three standard deviations.

From the laser ablation of each core to edge transect, six isotopes (magnesium-25, Mg; calcium-43, Ca; manganese-55, Mn; strontium-86, Sr; barium-138, Ba; and lead-208, Pb) were analyzed by ICP-MS. Counts per second were measured for each isotope along this transect, from which we calculated ratios of each isotope to calcium (e.g., Mg:Ca, mm mol−1) (Fig. 2B). Next, we calculated average isotope ratios for each period of summer and winter growth. Specifically, we used the otolith radius of each opaque and translucent zone (summer and winter growth, respectively) to designate the section of the core-to-edge transect for which to calculate the average for every isotope ratio (Fig. 2). When calculating the average isotopic ratio for the first summer of growth we excluded the yolk-sac stage because this stage has been shown in other species to be a product of maternal investment, and thus is not a site-specific signature (Brophy, Jeffries & Danilowicz, 2004; Chittaro et al., 2006). The yolk-sac stage was excluded from our isotope ratios by omitting the chemical signature of the laser transect segment within 100-µm of the otolith core. We were confident that this method excluded any maternal signal since yolk-sac absorption occurs 5–7 days after hatching (Bailey, 1982b) and 30–40 day old larvae are reported to have an otolith radius of 100–150 µm (Butler & Nishimoto, 1997).

Isotopes were included for statistical analyses if they met two criteria (Chittaro et al., 2013): concentrations of the isotope were greater than the detection limit in more than 80% of otoliths analyzed, and concentrations of the isotope in NIST samples were determined with satisfactory precision (coefficient of variation < 10%).

Generalized linear model

We used a generalized linear modeling (GLM) approach to investigate the extent to which variability in somatic growth (dependent variable) was explained by seven variables: year, harbor seal (Phoca vitulina) abundance, sea surface temperature, regional climate index, and several isotope ratios. This analysis was performed separately for each of three summers of growth, corresponding to the first, second, and third year of life. We used the glm function in the stats package of RStudio (R Core Team, 2021) and specified a gamma family distribution with a log link to account for the normally distributed, but positive, growth data.

We included year to account for interannual variation in marine environmental conditions not otherwise accounted for explicitly in the model. ‘Year’ corresponds to the year in which the summer growth occurred. For example, if a fish hatched in 1990 and we are investigating its second summer of growth, then the year associated with this growth would be 1992. Harbor seal abundance (Chasco et al., 2017) was included to account for the known predation pressure of harbor seals on Pacific hake (Gustafson et al., 2000). According to Saunders & McFarlane (1999), SoG hake represented 42% of the diets in harbor seals. Because Pacific hake are ectothermic their metabolic demands are positively correlated to water temperature. We therefore included sea surface temperature, averaged from April to August, collected from a buoy located at Race Rocks on the southeastern end of Victoria Island, British Columbia, Canada. To account for regional climate patterns we included Pacific Northwest Index (PNI) (Ebbesmeyer & Strickland 1995), which is a composite of location-specific air temperature, total precipitation, and snowpack data. Finally, because some aspects of water quality are recorded in the uptake of trace elements onto the growing otolith surface we used the chemical constituents of Pacific hake otoliths as a means to gain insight into the habitat where they resided during each summer. Specifically, we used average Mn:Ca as an indicator of hypoxic conditions (Limburg & Casini, 2018; Limburg et al., 2011; Limburg et al., 2015; Mohan & Walther, 2016), average Ba:Ca as an indicator of nutrient rich upwelling (Bath et al., 2000; Mohan et al., 2018), and average Sr:Ca (Macdonald & Crook, 2010; Martin, Thorrold & Jones, 2004; Zimmerman, 2005) and Ba:Ca (Elsdon & Gillanders, 2005; (Hamer, Jenkins & Coutin, 2006; Miller, Gray & Merz, 2010) as an indicator of salinity. Lead was excluded from our analyses because little is understood about how it relates to water quality.

We ran GLMs only for periods of summer growth and separately for each of the first three years, and each population. We limited our analyses to summer growth because yearly changes in growth were more dramatic, relative to winter, (see results) and harbor seal abundance data were collected in spring and summer (Chasco et al., 2017). The most complex model was describes as: G∼Yr+Seal+Temp+PNI+Mn+Sr+Ba

where G is the summer growth estimated from otoliths, Yr is the year the summer growth occurred, Seal is harbor seal abundance, Temp is the sea surface temperature, PNI is the Pacific Northwest Index, and Mn, Sr, and Ba are the average Mn:Ca, Sr:Ca, and Ba:Ca, respectively, corresponding to the summer of otolith growth (e.g., first summer of growth).

All possible GLM model combinations (n = 127) were run and model parameters were estimated by maximizing the likelihood function. To compare models we calculated four values for each model; Akaike’s Information Criterion (AIC), delta AIC, relative likelihood, and AIC weight. Smaller AIC values indicate “better” models and when comparing two models we calculated the difference in AIC values (delta AIC) (Akaike, 1973; Burnham & Anderson, 2002). A delta AIC of less than 2 indicates little difference between competing models; a delta AIC of 2–10 indicates moderate support for a difference between the models, and a delta AIC of greater than 10 indicates strong support (Burnham & Anderson, 2002). Relative likelihood represents the probability of a model given the data, whereas AIC weight is the discrete probability of each model (Burnham & Anderson, 2002).

The best model from each dataset was defined as having a delta AIC of 0.00, although preference was given to the model with the fewest variables if two or more models had a delta AIC of less than 2. This process was repeated for all six data sets: Puget Sound first, second, and third summer of growth and Strait of Georgia first, second, and third summer of growth.

Results

Age, length, and growth

We estimated the age of 439 Pacific hake collected from PS and SoG in 1979–2016 (Table 1). These fish ranged in age from young-of-the-year to 6, with 2 year-olds being most abundant and comprising 38% of all individuals (Fig. 3A). Based on these ages and when fish were collected, we estimated hatch year to range from 1974 to 2015 (Table 2). Median total length (mm) at capture was 243 mm and 410 mm for PS and SoG, respectively, with SoG fish showing a bi-modal size distribution (Fig. 3B).

Figure 3 Pacific hake age and size.

Frequency histogram of Pacific hake (A) age and (B) total length (mm) at capture from Puget Sound (white) and Strait of Georgia (black) populations. Grey color corresponds to overlap between populations.

Our reconstruction of Pacific hake length at each summer and winter together with our repeated measures ANOVA, across all hatch years, indicated a significant interaction between population and year/season of growth (F = 339.7; df = 1,1950; p = 0.00022). By the second summer, individuals from SoG were, on average, 18% larger than fish from PS (mean size of 265 mm and 215 mm, and standard deviation of 42 mm and 28 mm, respectively) (Fig. 4A). The repeated measures ANOVA comparing estimates of length per year/season of SoG fish did not reveal significant differences between those that hatched in the 1970s versus 1990s (Fig. 4B). However, the same analysis performed on PS fish revealed a significant interaction between decade in which they hatched and year/season (F = 40.6; df = 3,1201; p = 0.00022). Tukey’s post hoc test showed significant differences between fish that hatched in the 2010s to those that hatched in both 1980s and 1990s (Fig. 4C).

Figure 4 Otolith-derived estimates of total length at each year and season for Pacific hake collected from Strait of Georgia and Puget Sound.

Otolith-derived estimates of total length (mm) at each year and season (i.e., summer and winter) for Pacific hake collected from (A) Strait of Georgia and Puget Sound with pooled hatch years, and (B) Strait of Georgia and (C) Puget Sound fish grouped by the decade in which they hatched. Repeated measures ANOVA indicated significant length differences between fish from Puget Sound and Strait of Georgia (F = 303.2; df = 1,1952; p = 0.00022). Within populations, repeated measures ANOVA revealed only a significant interaction between decade and year/season for Puget Sound fish (F = 39.6; df = 3,1204; p = 0.00022). In plot C, lower case letters in the legend correspond to results from the Tukey’s post hoc test that revealed significant differences (at p < 0.05; indicated as different letters) between fish that hatched in the 1990’s and 2010’s. Black horizontal line, box, and whiskers represent median, first and third quartile (i.e., 25th and 75th percentiles), and minimum and maximum values, respectively. To improve visualization of the length differences among decades for Puget Sound fish (C), the maximum value on the y-axis was reduced (500 mm) relative to the other plots (600 mm).

Repeated measures ANOVA of our growth estimates, across all hatch years, indicated a significant interaction between population and year/season (F = 45.4; df = 1,1950; p = 0.00015). By the second summer individuals from SoG grew, on average, 42% more than fish from PS (mean growth 120 mm and 69 mm, and standard deviation of 28 mm and 31 mm, respectively) (Fig. 5A). The repeated measures ANOVA comparing estimates of growth per year/season of SoG fish did not reveal significant differences between those that hatched in 1970s versus 1990s (Fig. 5B). When we performed the same analysis on fish from PS it revealed a significant interaction (F = 15.7; df = 3,1201; p = 0.00031) between decade in which they hatched and year/season. Tukey’s post-hoc test showed significant differences among fish that hatched in the 1980s, 1990s, and 2010s, as well as between those that hatched in the 1980s and 2000s (Fig. 5C). The first and second summers showed the most dramatic increases and decreases, respectively, in growth across the decades in which fish hatched. Specifically, in their first summer fish that hatched in the 2010s grew, on average, 147 mm compared to 108 mm for those that hatched in the 1980s. However, fish that hatched in the 2010s grew 31 mm in their second summer while those that hatched in 1980s grew 107 mm (Fig. 5C).

Figure 5 Otolith-derived estimates of somatic growth at each year and season for Pacific hake collected from Strait of Georgia and Puget Sound.

Otolith-derived estimates of somatic growth (mm) at each year and season (i.e., summer and winter) for Pacific hake collected from (A) Strait of Georgia and Puget Sound with pooled hatch years, and (B) Strait of Georgia and (C) Puget Sound fish grouped by the decade in which they hatched. Repeated measures ANOVA indicated significant growth differences (F = 35.9; df = 1,1952; p = 0.00026) between fish from Puget Sound and Strait of Georgia. Within populations, repeated measures ANOVA indicated a significant interaction only between decade and year/season for Puget Sound fish (F = 16.9; df = 3,1204; p = 0.00054). In plot C, lower case letters in the legend correspond to results from the Tukey’s post hoc test that revealed significant pairwise differences (at p < 0.05; indicated as different letters) between fish that hatched among certain decades. Black horizontal line, box, and whiskers represent median, first and third quartile (i.e., 25th and 75th percentiles), and minimum and maximum values, respectively.

Generalized linear model

We obtained elemental concentrations from 214 individuals collected in PS and 132 from SoG. All isotopes analyzed were found in concentrations above the detection limit for 100% of the otoliths sampled. We observed an acceptable level of analytical precision based on NIST (n = 87) mean coefficient of variation and standard deviation: Mg:Ca (mean = 2.35, standard deviation = 1.44), Mn:Ca (mean = 2.42, standard deviation = 1.09), Sr:Ca (mean = 1.65, standard deviation = 0.52), and Ba:Ca (mean = 2.29, standard deviation = 1.13).

We investigated variability in somatic growth separately for each population owing to the significant differences in growth and relatively little overlap in hatch years between populations. Overall, our GLM approach to understand what factors explain variability in somatic growth rate revealed a different suite of variables, more or less, with each summer and population. Further, AIC weights were relatively low (<0.3) indicating that a substantial amount of variability in growth was unexplained.

For SoG fish, results of the GLM analysis indicated five models that explained the greatest amount of variability in the first summer of somatic growth and were indistinguishable (i.e., delta AIC < 2.0) (Table 3). Of these five models the simplest model was chosen as the best model and it showed a negative relationship between somatic growth and Mn:Ca and Sr:Ca, and a positive relationship between somatic growth and Ba:Ca, ‘Harbor seal abundance’, and ‘Sea surface temperature’ (Figs. 6A–6E). Our GLM analysis of the second summer of growth revealed nine models that best explained variability and were indistinguishable (Table 3). Of these nine models, the one with the lowest AIC was also the simplest model. This model showed positive and negative relationships between somatic growth and Mn:Ca and Sr:Ca, respectively (Figs. 6F–6G). Finally, the GLM analysis of the third summer of growth showed that nine models best explained variability and were indistinguishable (Table 3). Of these nine models, three each consisted of only one variable: ‘Sea surface temperature’, ‘Harbor seal abundance’, and ‘Year’. Each of these three models showed a negative relationship to somatic growth (Figs. 6H–6J).

Table 3 Results of the generalized linear modeling approach that assessed what variables explained variability in somatic growth within each of the first three summers, and within each population.

Relative likelihood (Like) is the likelihood of a model given the data, and AIC weight (AIC weight) is the discrete probability of each model. Only models that are indistinguishable (i.e., delta AIC (delta AIC) of ≤2.0) are displayed. Best models (X) are identified as models with the lowest delta AIC and with fewest variables. Population (Pop) corresponds to Strait of Georgia (SoG) and Puget Sound (PS).

Year	Pop	Model	Best	AIC	dAIC	Like	AICwt	
1	SoG	Year + Mn:Ca + Sr:Ca + Ba:Ca + Temp + PNI		1292.9	0.00	1.00	0.20	
1	SoG	Mn:Ca + Sr:Ca + Ba:Ca + Seals + Temp + PNI		1293.0	0.11	0.95	0.19	
1	SoG	Mn:Ca + Sr:Ca + Ba:Ca + Seals + Temp	X	1294.5	1.66	0.44	0.09	
1	SoG	Year + Mn:Ca + Sr:Ca + Ba:Ca + Seals + Temp		1294.7	1.80	0.41	0.08	
1	SoG	Year + Mn:Ca + Sr:Ca + Ba:Ca + Seals + Temp + PNI	 	1294.9	2.00	0.37	0.08	
2	SoG	Mn:Ca + Sr:Ca	X	1263.3	0.00	1.00	0.13	
2	SoG	Year + Mn:Ca + Sr:Ca + PNI		1264.4	1.10	0.58	0.07	
2	SoG	Mn:Ca + Sr:Ca + Ba:Ca		1264.7	1.40	0.50	0.06	
2	SoG	Mn:Ca + Sr:Ca + Temp		1265.1	1.87	0.39	0.05	
2	SoG	Year + Mn:Ca + Sr:Ca		1265.2	1.94	0.38	0.05	
2	SoG	Mn:Ca + Sr:Ca + Seals	 	1265.2	1.96	0.38	0.05	
3	SoG	Temp	X	815.7	0.00	1.00	0.06	
3	SoG	Temp + PNI		816.8	1.06	0.59	0.04	
3	SoG	Ba:Ca + Temp		817.1	1.39	0.50	0.03	
3	SoG	Seals + Temp		817.2	1.52	0.47	0.03	
3	SoG	Seals	X	817.3	1.60	0.45	0.03	
3	SoG	Year + Temp		817.3	1.60	0.45	0.03	
3	SoG	Year	X	817.4	1.66	0.44	0.03	
3	SoG	Sr:Ca + Temp		817.4	1.70	0.43	0.03	
3	SoG	Mn:Ca + Temp	 	817.6	1.87	0.39	0.02	
1	PS	Year + Mn:Ca + Sr:Ca + Ba:Ca + Seals	X	1946.1	0.00	1.00	0.30	
1	PS	Year + Mn:Ca + Sr:Ca + Ba:Ca + Seals + Temp		1947.2	1.12	0.57	0.17	
1	PS	Year + Mn:Ca + Sr:Ca + Ba:Ca + Seals + PNI	 	1947.9	1.82	0.40	0.12	
2	PS	Mn:Ca + Sr:Ca + Seals + Temp + PNI		1954.6	0.00	1.00	0.14	
2	PS	Mn:Ca + Sr:Ca + Seals	X	1955.4	0.80	0.67	0.09	
2	PS	Year + Mn:Ca + Sr:Ca + Seals + Temp + PNI		1955.6	0.99	0.61	0.09	
2	PS	Mn:Ca + Sr:Ca + Seals + Temp		1955.8	1.20	0.55	0.08	
2	PS	Mn:Ca + Sr:Ca + Ba:Ca + Seals + Temp + PNI		1956.0	1.37	0.50	0.07	
2	PS	Year + Mn:Ca + Sr:Ca + Seals	 	1956.4	1.75	0.42	0.06	
3	PS	Year + Mn:Ca + Ba:Ca + Seals	X	890.5	0.00	1.00	0.14	
3	PS	Year + Mn:Ca + Sr:Ca + Ba:Ca + Seals		892.1	1.65	0.44	0.06	
3	PS	Year + Mn:Ca + Ba:Ca + Seals + PNI		892.4	1.93	0.38	0.05	
3	PS	Mn:Ca + Ba:Ca + Seals + PNI	X	892.4	1.96	0.38	0.05	
3	PS	Year + Mn:Ca + Ba:Ca + Seals + Temp	 	892.5	1.98	0.37	0.05	

Figure 6 Model fits of Strait of Georgia somatic growth (mm) in the first (A–E), second (F–G), and third (H–J) summers with respect to predictor variables that best explained variability in somatic growth (Table 3).

Mn:Ca, Sr:Ca, and Ba:Ca are average values that correspond to the summer of otolith growth for the corresponding year (e.g., first summer of growth). 95 percent confidence intervals of the predicted values are represented as the dash lines.

With respect to PS fish, the GLM analysis of the first summer of growth showed that three models best explained variability in somatic growth and were indistinguishable (Table 3). Of these, the best was also the simplest. This model indicated positive relationships between somatic growth and year, Mn:Ca, Ba:Ca, and ‘Harbor seal abundance’, and negative relationships with Sr:Ca (Figs. 7A–7E). Results of the GLM analysis of the second summer of growth revealed that nine models best explained variability in somatic growth and were indistinguishable (Table 3). Of these, the best was also the simplest, in which a positive relationship was observed between somatic growth and Mn:Ca and a negative relationship to Sr:Ca and ‘Harbor seal abundance’ (Figs. 7F–7H). Lastly, the GLM analysis of the third summer of growth showed that seven models best explained variability and were indistinguishable (Table 3). Of these, two were the simplest model, both consisted of four variables, and had Mn:Ca, Ba:Ca, and ‘Harbor seal abundance’ in common (Table 3). Both models showed a positive relationship between growth and Mn:Ca, and negative relationships with Ba:Ca and ‘Harbor seal abundance’ (Figs. 7I–7L). In one model there was a negative relationship between growth and year, and a lack of a relationship with PNI.

Figure 7 Model fits of Puget Sound somatic growth (mm) in the first (A–E), second (F–H), and third (I–L) summers with respect to predictor variables that best explained variability in somatic growth (Table 3).

Mn:Ca, Sr:Ca, and Ba:Ca are average values that correspond to the summer of otolith growth for the corresponding year (e.g., first summer of growth). 95 percent confidence intervals of the predicted values are represented as the dash lines.

Discussion

The ability of a population to be sustainably managed depends upon its rate of harvest and recovery. With respect to Pacific hake in the Georgia Basin, there has been two different responses to harvest with one population failing to recover despite decades of a closed fishery. In this study we evaluated what factors and processes may be involved in preventing Pacific hake recovery by examining otolith-derived estimates of somatic growth across several decades. We observed dramatic changes in seasonal growth and size-at-age across sampling years, especially for fish sampled from PS.

Size-at-age and growth differences between populations

Our findings of size-at-age differences between PS and SoG populations (Fig. 4A) were expected given that the listing of the Georgia Basin DPS as a Species of Concern was due, in part, to the reduced size-at-age of PS fish observed through the 1990s (Gustafson et al., 2000). For example, Pedersen (1985) and Goñi (1988) reported that PS fish collected in the mid-1960s to early 1980s grew approximately 2–5 cm shorter than those from SoG. What our study also revealed is that this divergence in body size between fish from PS and SoG occurred during their second summer of life (Fig. 4A). Size differences further increased by their third summer when PS fish grew, on average, 73% less than fish from SoG (i.e., 23.2 mm compared to 88.7 mm, respectively; Fig. 5A). It should be noted, however, the limited temporal overlap of fish that hatched in PS and SoG. Specifically, only the hatch years from 1994 through 1998 included individuals that were sampled from both PS and SoG (Table 2).

From 1965–1974 most of the PS fish ranged in size from 32 cm to 45 cm (average length of 36.2 cm, with recorded maximum lengths of 45 cm and 73 cm for males and females, respectively) (Pedersen, 1985), yet since 2008, less than 5% of sampled PS fish (n = 11,652) had a length greater than 30 cm (Blaine, Lowry & Pacunski, 2020) and Supplemental Table 1). Pedersen (1985) suggested that the declining body size in PS through time, and the fact that PS fish mature at a smaller size, relative to SoG fish, may have been due to the intense commercial fishery in PS in the 1970s and 1980s. Specifically, the extensive removal of large individuals from a population has been shown to select for individuals that mature earlier and grow less (Allendorf & Hard, 2009; Olsen et al., 2004; Sharpe & Hendry, 2009). For instance, California sheephead (Semicossyphus pulcher) had significantly smaller body size, earlier maturation, and a reduced maximum lifespan following more than 20 years of high fishing pressure (Hamilton et al., 2007). Similarly, zebrafish (Danio rerio) invested more in reproduction and attained a smaller adult body size when exposed to intense size-selective harvesting (Uusi-Heikkilä et al., 2015). Interestingly, models simulating zebrafish population dynamics indicated a slow population recovery even after a moratorium on harvest. Uusi-Heikkilä et al. (2015) suggested that this lack of population recovery was linked to changes in body size and reproductive investment, which placed the population at a disadvantage to manage natural selection pressures that often favor large body size. Similar harvest-related changes to life-history could explain why the smaller body size and the lack of recovery of the PS population persists today despite the closure of the fishery for thirty years.

Size-at-age and growth differences through time and within populations

We observed significant differences in size-at-age of PS fish such that a shift was seen towards smaller fish through time (Fig. 4C), yet no differences in size-at-age were detected between SoG fish that hatched in the 1970s and 1990s (Fig. 4B). In terms of somatic growth, we found a similar pattern in both populations whereby in recent years fish tended to grow more in their first summer and less in subsequent summers. Specifically, SoG fish that hatched in the 1990s grew, on average, 11% more in the first summer, and 25% less in the third summer, relative to those that hatched in the 1970s (Fig. 5B). In comparison, PS fish that hatched in 2010s grew 26% more in their first summer than those that hatched in 1980s, yet 71% less in the second summer compared to those that hatched in 1980s (Fig. 5C). Beamish & McFarlane (1999) suggest that this increased somatic growth during the first summer is linked to a shift in climate and ocean conditions that occurred in the late 1980′s. This regime shift is hypothesized to have caused a greater temporal overlap between young-of-year hake and an important prey item (e.g., copepods) in SoG (King & McFarlane, 2006). Because PS and SoG populations are part of the same basin and share similar characteristics it is possible that the same processes played a role in the size and growth patterns in SoG would be seen in PS.

Our observed decrease in growth during the third summer of SoG fish is consistent with findings by King & McFarlane (2006) whereby an increase in hake abundance following the aforementioned regime shift likely lead to reduced somatic growth later in life because of increased density dependent competition. The third year of growth is an important period in hake development in which they become reproductively mature and are suspected to move deeper to feed (Saunders & McFarlane, 1999). Therefore, if intraspecific competition is high during the time when surplus energy is allocated to reproductive development then it is expected that reduced energy will be available for growth. Regardless of the mechanism(s), the declines in growth through time, in both populations, highlights increased concern for the sustainability of this Species of Concern.

Our investigation of what factors and/or processes explain variability in seasonal growth indicated some similarities between populations. For instance, growth in the first summer was best explained by models that included positive relationships to harbor seal abundance and to barium levels (i.e., Ba:Ca) (Table 4). The positive growth associated with harbor seal abundance could indicate that hake resources are density-mediated and therefore predation-induced reductions in hake abundance may have led to reductions in intraspecific competition (Heithaus et al., 2008). Ba:Ca is positively associated with areas of nutrient rich upwelling (Mohan et al., 2018; Wheeler et al., 2016) and freshwater input (Elsdon & Gillanders, 2005; Hamer, Jenkins & Coutin, 2006; Miller, Gray & Merz, 2010), and the positive relationship between Ba:Ca and growth (Figs. 6C and 7D) might be due to increases in productivity, which could have increased prey (e.g., copepods) abundance due to nutrient enriched waters brought to the surface and/or from riverine sources. For instance, Takahashi et al. (2012) reported a positive relationship between otolith growth of northern anchovy (Engraulis mordax) and upwelling intensity, which they attributed to increased productivity and in turn improved prey nutritional value. Strontium (i.e., Sr:Ca) in otoliths is reported to be positively associated with salinity (Macdonald & Crook, 2010; Martin & Thorrold, 2005; Zimmerman, 2005). Thus the negative relationship we observed in both populations between growth (in the first and second summers) and Sr:Ca (Table 4; Figs. 6B, 6G, 7C, and 7G), and the aforementioned relationship to Ba:Ca, suggests that the first two years of growth are sensitive to freshwater flow, such that growth declines with a reduction in freshwater. Main spawning aggregations for both PS and SoG populations occur in close proximity to major sources of freshwater (e.g., Stillaguamish and Snohomish rivers in PS and Fraser River in SoG). This freshwater input would cause a stratified layer of low-salinity water above the well-mixed marine layer. This is significant since Pacific hake larvae are known to aggregate in surface waters (Gustafson et al., 2000).

Table 4 Results of the GLM analysis that identified independent variables that best explained variability in summer growth.

Positive (+) and negative (-) relationships, and lack of a relationship (ϕ) are indicated for each population (Strait of Georgia, SoG, and Puget Sound, PS) with respect to each of three summers. The relationships indicated in the first and second summers correspond to the variables found in the model that best explains variability in growth for both populations (see Table 3). For the third summer for SoG, each of the three variables represents its own model. For the third summer for PS, two models each of which consisted of four variables and had three variables in common (Mn:Ca, Ba:Ca, and Seals) and differed in the fourth variable (either PNI or Year).

	1st summer	2nd summer	3rd summer	
Variable	SoG	PS	SoG	PS	SoG	PS	
Mn:Ca	–	+	+	+		+	
Sr:Ca	–	–	–	–			
Ba:Ca	+	+				-	
Seals	+	+		–	–	-	
Year		+			–	-	
SST	+				–		
PNI						∅	

The third summer of growth for fish from both populations was negatively associated with harbor seal abundance, which is an opposite pattern observed for the first summer of growth. These positive and negative relationships may suggest that younger hake experience a competitive release in years with more harbor seals, while older hake experience reduced performance possibly due to energy expenditure through predatory avoidance. Heithaus et al. (2008) noted that aside from direct mortality, predators can strongly influence prey behavior such as foraging. It is likely that hake recovery, particularly in PS, is limited given that harbor seal abundances increased by 7% (Saunders & McFarlane, 1999) and 3% (Gustafson et al., 2000) in SoG and PS, respectively, in the early to mid-1990s and appear to be stable in both populations (Jeffries et al., 2003; Olesiuk, 1999).

Conclusion

Declines in fish body size have been reported in many commercial stocks (Ohlberger et al., 2018; Sharpe & Hendry, 2009) and these changes likely have important ramifications for the sustainability of the harvested species (Allendorf & Hard, 2009; Losee, Kendall & Dufault, 2019) as well as ecosystem function and services (Oke et al., 2020). The Pacific hake Georgia Basin DPS are prey to top predators and are important to a healthy and functioning ecosystem (Gustafson et al., 2000). Unfortunately, the Georgia Basin DPS has experienced substantial changes in juvenile somatic growth over the last several decades, despite fishery regulations, which highlights increased concern for the sustainability of this Species of Concern. Further, our investigation of juvenile somatic growth indicated that a diverse suite of processes (e.g., competition and predation) and factors (e.g., salinity and upwelling) were important for explaining variability and that these relationships differed with respect to population and age. These results highlight the complicated nature of population recovery under dynamic, and typically uncontrollable, variation in biotic and abiotic conditions.

Supplemental Information

Supplemental Information 1 Relationship between total length (mm) at capture and otolith radius (microns) for Pacific hake collected from Strait of Georgia (blue circles) and Puget Sound (red circles) populations

Biological intercepts of length and otolith radius at first feeding were defined as 3.9 mm and 14.1 µm, respectively

Click here for additional data file.

Supplemental Information 2 Length frequency data of Pacific hake collected from Puget Sound from 1987–2021

Data provided by Washington Department of Fish and Wildlife. For more details about the Puget Sound survey see Blaine J, Lowry D, Pacunski R (2020) 2002–2007 WDFW scientific bottom trawl surveys in the southern Salish Sea: species distributions, abundance, and population trends. Washington Department of Fish and Wildlife, Olympia, WA. Pp. 252. https://wdfw.wa.gov/sites/default/files/publications/02140/wdfw02140.pdf

Click here for additional data file.

Supplemental Information 3 Raw data associated with each Pacific hake collected by Department of Fisheries and Oceans, Canada & Washington Department of Fish and Wildlife

Raw data of Pacific hake location and date of collection, age and size of fish, total otolith radius and otolith radius at each opaque and translucent zones (which correspond to summer and winter otolith growth), average trace element ratios, abundance of hake, harbor seal, and orca, climate data (e.g., PDO, ONI, upwelling, NPI, PNI), salinity, and sea surface temperature.

Click here for additional data file.

Special thanks to J Blaine, S Richardson, K Sobocinski, N Tolimieri, M Sorel, B Nelson, and E Ward who provided data and helped with analyses and to R Gustafson and E Iwamoto for their constructive comments. Age reading of otoliths was provided by staff of the Washington Department of Fish and Wildlife and Department of Fisheries and Ocean ageing labs. We also thank C Knaak, C Fisher, and J Vervoort at the GeoAnalytical Lab, Washington State University for help operating the mass spectrometers.

Additional Information and Declarations

Competing Interests

Author Contributions

Animal Ethics

Field Study Permissions

Data Availability

1 ESA allows for conservation of organisms at the subspecies level and thus defines “species” as “any subspecies of fish or wildlife or plants, and any distinct population segment of any species of vertebrate fish or wildlife which interbreeds when mature” USFWS-NMFS (1996) Policy regarding the recognition of distinct vertebrate population segments under the Endangered Species Act. Federal Register. 61(26):4722-4725.

2 Reference to trade names does not imply endorsement by the National Marine Fisheries Service, NOAA.

The authors declare there are no competing interests.

Paul Chittaro conceived and designed the experiments, performed the experiments, analyzed the data, prepared figures and/or tables, authored or reviewed drafts of the article, and approved the final draft.

Chris Grandin performed the experiments, authored or reviewed drafts of the article, collected and aged fish, and approved the final draft.

Robert Pacunski performed the experiments, authored or reviewed drafts of the article, collected and aged fish, and approved the final draft.

Rich Zabel analyzed the data, authored or reviewed drafts of the article, and approved the final draft.

The following information was supplied relating to ethical approvals (i.e., approving body and any reference numbers):

Animal ethics and care approval was not required for this study because otoliths were collected as part of other studies.

The following information was supplied relating to field study approvals (i.e., approving body and any reference numbers):

Field work was conducted by Washington Department of Fish and Wildlife and Department of Fisheries and Ocean Canada.

The following information was supplied regarding data availability:

The raw data measurements of Pacific hake are available in the Supplementary File.

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
