# Peer review of "Five decades of change in somatic growth of Pacific hake from Puget Sound and Strait of Georgia"

_PeerJ, doi:10.7717/peerj.13577_

## Round 0.1 · original submission · Major Revisions

As you can see, the manuscript has been commented on by two reviewers, suggesting a moderate number of changes. In general, however, it has been received positively, and the comments and the clarification requests are all reasonable and represent a chance to further improve the paper. I ask you, in particular, to pay particular attention to the comments related to the choice of the back-calculation method, questioned by both reviewers. A very specific minor comment from my side: was the relevance of the variables in table 4 (GLMs) assessed on the basis of the AIC-based model selection alone? If so, I would avoid the use of the term “significant” in the table and in the text, as it would suggests that formal tests of significance were performed.

Reviewer 1 ·

Basic reporting

The manuscript is adequately presented and clearly structured. The different sections are easy to follow and coherent. The figures and tables are also clear and nicely presented. Raw data is available for checking and seems suitable.

Experimental design

The research question and framing of the introduction is well-crafted and sums up well the background, theory and expected contribution behind this analysis, particularly the changes in growth rate and their timing in the different populations. I have however a few concerns and/or questions (some of which are detailed in the next section).

First, being more familiar with its Atlantic counterpart, how accurate is Pacific hake aging using otolith increments, and has there been any validation experiments done prior? I’m asking since European hake aging using otoliths has recently become more and more problematic and even removed from current stock assessment in favor of Age-Length Keys. The main issue was a consistent inter-reader overestimation / underestimation when looking at increments over 3 years old, and an overall zonation pattern proven to be less straightforward than initially thought. In the present manuscript the various source used to reference otolith aging of Pacific hake are old and correspond to the onset of similar work on European hake before it was refuted.

I’m also curious about some of the methodological choices made for the growth back-calculation using otolith increments. Why wasn’t the biological intercept method modified from Fraser Lee used instead of the base one? In terms of results it consistently performs equally or better than the FL method, but has the advantage of basing it on real measurements so that the emergence of proportionality between fish and otolith growth is correctly estimated. The issue with FL is that it uses a statistically determined intercept based on the relationship between otolith and fish length (showed in Appendix 1), which is directly dependent on the samples available and in particular should be treated carefully for growth estimates beyond the age classes represented in the data.

Here there is a reasonable number of samples with a good R2 but the populations are not equal. For example, the Strait of Georgia population stops at age 2 when it comes to available samples, yet a large portion of this manuscript discusses growth during first summer. Linear proportionality between fish and otolith growth is known to be particularly unreliable in the younger ages, especially when those are only derived from the back-calculation model. I’d be careful with including 1st summer back calculations for SoG, or at least I’d like to see more mentions of this potential limitation either in methods or in the discussion.

Validity of the findings

The discussion did a good job to provide a summary of the results and recontextualize them with the research questions. A clear and concise conclusion is also provided at the end to summarize the key findings, encompassing both the modelling results on the factors driving growth dynamics. Overall, it is a good quality work but there remains some issues.

I don’t have much to say about the first part of the analysis. The multivariate analysis is suitable and well presented, and shows interesting temporal and inter-populations differences. I am somewhat conflicted on other aspects and in particular the following modelling approach.

First, regarding the discussion of trace elements in the growth modelling. Ba signals have been identified in nutrient-rich waters, and given the particular dynamic of SoG and PS there is certainly a high likelihood for upwelling induced chemical signals that positively correlate with growth. The authors discuss the possible positive effect of freshwater through the negative correlation between growth and Sr, but it would have been good to discuss it in light of the Ba signal too, given that in both SoG and PS there is a strong dependence on freshwater influx from the nearby rivers for bringing in nutrients.

Second, the analysis could have benefited from more consideration of the fishing factors. The idea of fishery-induced physiological changes, notably an earlier maturation and smaller / growth, is adequately brought in the discussion, but lacking from direct analysis. In both populations the authors show a difficulty separating the different models due to low delta AICs, and a generally large proportion of unexplained variance. It would have been interesting to test those models after including some form of fishery-associated factors, for example F. I suspect they would have been easier to separate and would have added stronger signals. In turn, it could have also become interesting to contrast the influence of harbour seal predation on density-dependent growth with fishing-induced density release.

Additional comments

There is very little mention of limitations throughout the manuscript, either experimental or methodological. The GLM results reported lack statements on uncertainty and confidence, if would be good to report at least CIs in the different plots from figure 6 and 7.

A model equation should be included in the method section for clarity (I suggest the most complex model as reference). Out of curiosity, did the authors consider a GLMM (with random effects) instead? Given the natural variability in individual growth trajectories, it is usually quite important to include at least a fish-specific random intercept.

Regarding elemental values, there is mention of averaging them but no further details. Were those averaged per age? Per summer zone? Over the whole otolith? It should be specified in the text as well as in the legend of figures 6 and 7.

L.126 : add Lc = length at capture to the equation description

Table 1: add a divider between the two population as it can be confusing to read from SoG age 6 to PS age 0. Eventually a sub-total for each pop.

Figures 6 and 7: add confidence intervals

Reviewer 2 ·

Basic reporting

Review of the manuscript peerj-69048


Manuscript tittle: “Five decades of change in somatic growth of Pacific hake from Puget Sound and Strait of Georgia”

Chittaro et al. wrote an interesting manuscript comparing the somatic growth of two populations of the Pacific hake in North America. They use different descriptors (predator abundance, otolith microchemistry and temperature proxies) to understand the drivers of differential somatic growth between the populations and through time within the populations. The dataset is robust and the statistical analysis are sound. Therefore, the authors presented interesting results and discussed them properly. I believe that the manuscript can be accepted for publications in PeerJ after minor revisions. Specific comments by section presented below:

Lines 43 to 44 – It is not usual to start the introduction with a question. I recommend the authors to rephrase this sentence.

Line 126 – Why did the authors used Fraser-Lee method instead of more recent and potentially more precise methods for back-calculations from otoliths? See Vigliola & Meekan 2009 (The Back-calculation of fish growth from otoliths).
Lines 173 to 174 – It would be better to have the chemical symbol of the elements instead of their names.
Line 235 – The authors could cite the R packages they used, once Rstudio it is just an UI from R software.

Figures 4 and 5 should use colors that are colorblind friendly. Also, I wonder if wouldn’t be clearer to have the information of winter and summer presented differently, because the somatic growth during winter is much smaller than during summer, so it is harder to identify changes in growth for winter when presented in the same axis as the summer values.

Experimental design

The experimental design is good and the research questions are well defined and the methods are also clearly described.

Validity of the findings

The results are interesting and have broad impact (it can be used as a comparison to other fish populations elsewhere). The statistical analysis were robust and the interpretation of the results well discussed.

---

## Round 0.2 · accepted · Accept

As you can see the reviewers appreciated the work done to improve the manuscript and the attention given to each comment they raised on the previous version. I agree with them on the improvements and on the fact that the manuscript is now ready to be published

Reviewer 1 ·

Basic reporting

No additional comment from the initial review. Overall, I commend the authors for the clear and thorough review work on their manuscript. I appreciate the detailed answers and considerations given to each point, and the incorporation of most of the more important comments within this reviewed manuscript. There isn't much to be added, though I'd still like to voice my concerns about aging of Merluccius species using otoliths (see next section).

Experimental design

The inclusion of the biological intercept has solved most of my issues with the otolith growth model. I am thankful for the addition of a growth equation, as well as the integration of Ba in the chemistry results.

While I appreciate the technical reports sent as support for an otolith-based individual aging used in assessment, I am still somewhat skeptical of its validity in an hake species given that there haven't been modern validation experiments to back it up beyond the 1985 paper provided here. Indirect validation through tracking of cohorts is a fine approach, but it is sensitive to biases when human interpretation is straightforward but have no biological meaning. Here the analysis relies on estimates of yearly growth based on the interpretation of annual rings: my point is that without any direct experiment of otolith deposition, the ages and subsequent growth estimates could be wrong

The European hake was for a long time assessed similarly, and issues with aging only arose in the last two decades when it was realized the deposition pattern was less straightforward than anticipated, and age estimates were consistently wrong especially in young fish. I therefore recommend the authors to be careful of any aging done on the Merluccius genus, and alternatively would encourage new validation experiments to see if aging of Pacific hake using otolith is indeed biologically plausible.

Validity of the findings

No additional comment

Additional comments

No additional comment

Reviewer 2 ·

Basic reporting

Chittaro et al. amended the all the issues pointed by both reviewers and the editor, therefore I believe that the manuscript can be accepted for publication in Peerj.

Experimental design

The manuscript was well designed and analyzed. The results are sound and they were interpreted correctly.

Validity of the findings

The authors reported an interesting case of analysis in fish somatic growth, reporting data on growth during winter season, which is not usually done. They findings can be useful for the scientific community.